# Genetic identification of bat species for pathogen surveillance across France

Youssef Arnaout[1,2], Zouheira Djelouadji[2], Emmanuelle Robardet[1], Julien Cappelle[3,4], Florence Cliquet[1], Frédéric Touzalin[5], Giacomo Jimenez[6], Suzel Hurstel[7,8], Christophe Borel[6], Evelyne Picard-Meyer[1] *

1 ANSES-Nancy Laboratory for Rabies and Wildlife, WHO Collaborating Centre for Research and Management in Zoonoses Control, OIE Reference Laboratory for Rabies, European Union Reference Laboratory for Rabies, European Union Reference Laboratory for Rabies Serology, Malzéville, France, 2 VetAgro Sup Lyon Laboratory for Leptospira, Marcy l'Etoile, France, 3 UMR ASTRE, CIRAD, INRAE, Université de Montpellier, Montpellier, France, 4 UMR EPIA, INRAE, VetAgro Sup, Theix, France, 5 School of Biology and Environmental Science, Science Centre West, University College Dublin, Dublin, Ireland, 6 CPEPESC Lorraine, Neuves-Maisons, France, 7 GEPMA, Strasbourg, France, 8 LPO Alsace, Rosenwiller, France

* evelyne.picard-meyer@anses.fr

**Data Availability Statement:** All files are available from the Genbank database (accession number(s) MZ066766 to MZ066790). All relevant data are

## Abstract

With more than 1400 chiropteran species identified to date, bats comprise one-fifth of all mammalian species worldwide. Many studies have associated viral zoonoses with 45 different species of bats in the EU, which cluster within 5 families of bats. For example, the Serotine bats are infected by European Bat 1 Lyssavirus throughout Europe while Myotis bats are shown infected by coronavirus, herpesvirus and paramyxovirus. Correct host species identification is important to increase our knowledge of the ecology and evolutionary pattern of bat viruses in the EU. Bat species identification is commonly determined using morphological keys. Morphological determination of bat species from bat carcasses can be limited in some cases, due to the state of decomposition or nearly indistinguishable morphological features in juvenile bats and can lead to misidentifications. The overall objective of our study was to identify insectivorous bat species using molecular biology tools with the amplification of the partial cytochrome b gene of mitochondrial DNA. Two types of samples were tested in this study, bat wing punches and bat faeces. A total of 163 bat wing punches representing 22 species, and 31 faecal pellets representing 7 species were included in the study. From the 163 bat wing punches tested, a total of 159 were genetically identified from amplification of the partial cyt b gene. All 31 faecal pellets were genetically identified based on the cyt b gene. A comparison between morphological and genetic determination showed 21 misidentifications from the 163 wing punches, representing ~12.5% of misidentifications of morphological determination compared with the genetic method, across 11 species. In addition, genetic determination allowed the identification of 24 out of 25 morphologically non-determined bat samples. Our findings demonstrate the importance of a genetic approach as an efficient and reliable method to identify bat species precisely.

within the manuscript and its Supporting information files.

**Funding:** This work was supported by the French National Agency for Food, Environmental and Occupational Health & Safety and VetAgro Sup laboratory for Leptospira.

**Competing interests:** No authors have competing interests.

**Abbreviations:** Cox, Cytochrome c oxidase subunit 1; Cyt b, Cytochrome b; DNA, Deoxyribonucleic acid; LPO, League for the Protection of Birds; LRFSN, Nancy Laboratory for Rabies and Wildlife; mt, Mitochondrial; n.d., Not determined; PCR, Polymerase Chain Reaction.

# Introduction

All bat species and their roosts are legally protected in France and in Europe by national and international legislation due to the significant decrease in their populations over the last few decades. Bats belong to order Chiroptera, the second largest order of mammals after rodents. With more than 1400 chiropteran species identified to date, bats comprise one-fifth of all mammalian species worldwide. Bats are divided into two sub-orders: Yangochiroptera including 12 microbat families, and Yinpterochiroptera including four microbat families in Rhinolophidea plus Old World fruit bats [1, 2]. Today, 51 bat species have been identified in Europe of which 35 occur in France, including one new cryptic species (*Myotis crypticus*) recently identified in France [3, 4]. All bats in Europe are insectivorous except *Rousettus aegyptiacus*, which is a frugivorous bat commonly reported in Africa, but also in Cyprus in Europe [5, 6]. Insectivorous bats in Europe are divided into four different families: Rhinolophidae, Vespertilionidae, Molossidae and Miniopteridae [7, 8]. Some studies have suggested a possible association between some bat species and micro-organisms, including more than 200 viruses, bacteria, parasites and pathogenic fungi. For example, *Myotis daubentonii*, found throughout Ireland and Europe and as far as Japan and Korea, has been shown to be infected with alphacoronaviruses, astroviruses, paramyxoviruses [9, 10], lyssaviruses [11] and *Bartonella* bacteria [12]. *Myotis myotis*, common in France, has been reported with the presence of alphacoronaviruses [13], herpesviruses and with *Yersinia* bacteria [14].

The role of bats in the transmission of zoonotic pathogens to both humans and animals is not clear. Their possible specific association with infectious micro-organisms highlights the importance of identifying bat species precisely to thoroughly investigate the link between the potential presence of pathogens and bat species. Species identification of individuals is generally performed by bat specialists using morphological keys [15]. The morphological determination of bats is usually based on geometric morphometrics since the 1980s [16–18]. For example, since 1989, to differentiate between *M. myotis* and *M. blythii* populations, each individual has been identified with external measurements such as left and right forearm lengths, third- and fifth-digit lengths, ear width, ear length, weight, ear surface area, tragus form, number of ear folds, calcar tip form, fur colour. This set of techniques has become popular and has been used for determining bats across a large number of taxa. In some cases, morphological determination can lead to misidentification of some bat species, especially from sites harbouring multiple species belonging to the same family [19]. Moreover, it is extremely difficult to determine some species precisely based on carcasses, for example *Pipistrellus pipistrellus*, *P. kuhlii*, *P. pygmaeus* and *P. nathusii*, or to distinguish between young *P. pipistrellus* and *P. pygmaeus* due to their similar morphological features. Finally, differences may be less obvious between these species when carcasses are in an advanced state of decomposition.

Many studies have shown the importance of genetic determination by amplification of mitochondrial DNA (mtDNA) for accurate identification of bat species [20–22]. Animal mtDNA is generally a small, circular molecule containing 37 genes that are essential for normal mitochondrion function [23]. Species identification and DNA barcoding have been shown to be a useful tool to better understand the relationship between the presence of pathogens and bat species. For instance, a study showed the role of bat species involved in the circulation of lyssaviruses across Canada [19, 20]. Genetic identification can be undertaken by testing for the partial cytochrome b gene (cyt b) [24], cytochrome c oxidase subunit 1 (cox1) gene [20] or a portion of the hypervariable domain II of the mtDNA control D-loop region that can differentiate mitochondrial haplotypes and diversity [25, 26].

Interestingly, new universal cyt b primers allow species identification of 63 animal species belonging to 38 families from 14 orders and 5 classes (Mammalia, Aves, Reptilia,

Actinopterygii and Malacostraca) from putrefied samples [27]. Primer design is based on an alignment of referenced cyt b gene sequences (-1140 nt) from 751 Mammalia species, including bats. Primers have been used for the identification of different animal species belonging to 38 families, except bats. Many types of sample have been tested, including muscle, brain, lung or spleen tissue, blood, oral swabs, and others [20, 27]. However, the drawbacks of collecting these types of samples involve the need to capture and restrain the animals combined with the difficulty of handling them. To avoid sampling live animals, using a non-invasive sampling technique such as faeces sampling can be an alternative solution to the capture of bats. Faecal samples represent a simple and easy method to collect samples from living bats without disturbing them using capture/release methods [28, 29].

One study has demonstrated the possibility of genetically identifying bat species from guano samples and other non-invasive samples based on the amplification of a segment of the mitochondrial gene cox1 [21]. Despite the fact that some studies have shown disadvantages of studying faeces samples, due to the presence of PCR inhibitors, fragmented DNA and the poor quality of extracted nucleic acids [30], other studies have demonstrated the efficacy and success of studying bat guano [9, 21].

The aim of this study was 1) to optimize the rapid PCR method previously described in Lopez-Oceja et al. (2016) with the new universal cyt b primers to identify autochthonous bat species from different types of bat sample, namely guano and wing punches tested for the first time; 2) to genetically determine bats in France and 3) to compare the morphological and genetic species identification of bat carcasses submitted for rabies diagnosis in 2018 and 2019.

## Materials and methods

### Bat specimens

The specimens used in this study were selected from a frozen and archived collection of bat carcasses submitted to the ANSES-Nancy Laboratory for Rabies and Wildlife for rabies diagnosis between 2018 and 2019. Wing punches (each ~8 mm, ~ 0.02 mg) were sampled from bat carcasses diagnosed negative for rabies and stored at -20˚C. All bats were previously identified using a morphological identification key by bat specialists [15]. The choice of bat samples was based on the following essential criteria: bat species and the geographic zone of collection. A total of 200 bat carcasses belonging to one of three families, Rhinolophidae, Vespertilionidae and Miniopteridae, representing 22 species were included in the study. Of the 200 bat wing punches tested, 37 were included in the development of the PCR and 163 were used in the PCR amplification of the partial cyt b gene followed by sequencing of amplified products and sequence analysis. Tables 1 and 2 gives the characteristics of the 200 bat specimens used in this study.

In addition, bat guano (one faecal pellet ~50 mm$^2$; ~ 0.02 mg) was also collected by bat specialists from the French Bird Protection League (LPO) Alsace as part of authorized bat studies. Faecal pellets were collected directly on the ground under the bat colony in three different sites in the Grand Est region in France. Bat species were determined in each selected area by inspected hanging individuals in the colony. A total of 31 bat faecal samples representing 7 species belonging to the families Rhinolophidae and Vespertilionidae were included in the genetic identification study (Table 3). Samples were collected in individual bags, stored at -20˚C and then at -80˚C before analysis.

### Ethics statement

Bats are protected species in Europe and in France. All biological samples employed in this study had been submitted for rabies diagnosis by ANSES-Nancy Laboratory for Rabies and

**Table 1. Characteristics of the 163 bat carcasses included for genetic identification.**

| Family | Bat species* | Number wing punches tested | | Total |
|---|---|---|---|---|
| | | 2018 | 2019 | 2018–2019 |
| Vespertilionidae | *Barbastella barbastellus* | 1 | | 1 |
| | *Eptesicus serotinus* | 1 | 7 | 8 |
| | *Eptesicus nilssonii* | 1 | | 1 |
| | *Myotis bechsteinii* | 1 | | 1 |
| | *Myotis brandtii* | | 1 | 1 |
| | *Myotis daubentonii* | | 3 | 3 |
| | *Myotis emarginatus* | 1 | | 1 |
| | *Myotis myotis* | | 11 | 11 |
| | *Myotis mystacinus* | 1 | 4 | 5 |
| | *Myotis nattereri* | | 3 | 3 |
| | *Nyctalus leisleri* | 3 | 10 | 13 |
| | *Nyctalus noctula* | 2 | 8 | 10 |
| | *Pipistrellus kuhlii* | 3 | 4 | 7 |
| | *Pipistrellus nathusii* | 2 | 15 | 17 |
| | *Pipistrellus pipistrellus* | 1 | 24 | 25 |
| | *Pipistrellus pygmaeus* | 2 | 2 | 4 |
| | *Plecotus* | | 1 | 1 |
| | *Plecotus auritus* | 2 | 5 | 7 |
| | *Plecotus austriacus* | 3 | 6 | 9 |
| | *Vespertilio murinus* | | 2 | 2 |
| | *Pipistrellus* sp. | 3 | 5 | 8 |
| | n.d. | | 18 | 18 |
| Rhinolophidae | *Rhinolophus ferrumequinum* | | 2 | 2 |
| | *Rhinolophus hipposideros* | | 4 | 4 |
| Miniopteridae | *Miniopterus schreibersii* | | 1 | 1 |
| | Total of samples tested | 27 | 136 | 163 |

n.d.: not determined.

*: identification based on morphological criteria.

Wildlife in accordance with the formal authorization by the French Ministry of the Environment [31]. In France and within the European Union, the legal frame- work for using under experimentation purposes is governed by Regulation 2010/63/EU of the European parliament and of the council of 22 September 2010 (applicable and translated in French in 2013) and handling of wildlife animal in the field does not require any prior specific ethical approval.

## DNA extraction

DNA extraction was performed using 1 punch per animal or 1 faecal pellet per site or per bat. Wing punches were directly used for DNA extraction, whereas a pre-extraction step was carried out to prepare bat faeces. Each faecal pellet was ground with 120 μL of 1X PBS buffer (phosphate buffered Saline, Sigma-Aldrich, Saint Quentin-Fallavier, France) then centrifuged for 5 min at 30,000 x g. For DNA extraction, 20 μL of supernatant was used and the extraction was performed using the Nucleospin Tissue Kit (Macherey Nagel, Hoerdt, France), following the manufacturer's recommendations. DNA samples were quantified using a Qubit fluorometer (Invitrogen, Marseille, France) and stored at -20˚C before use.

**Table 2. Characteristics of the 37 bat carcasses included for optimization.**

| Family | Bat species* | Number wing punches tested | | Total |
|---|---|---|---|---|
| | | 2018 | 2019 | 2018–2019 |
| Vespertilionidae | *Barbastella barbastellus* | | 1 | 1 |
| | *Eptesicus serotinus* | 1 | 1 | 2 |
| | *Eptesicus nilssonii* | | 1 | 1 |
| | *Myotis daubentonii* | | 1 | 1 |
| | *Myotis emarginatus* | 1 | | 1 |
| | *Myotis myotis* | 2 | | 2 |
| | *Myotis mystacinus* | | 2 | 2 |
| | *Myotis nattereri* | | 1 | 1 |
| | *Nyctalus leisleri* | 1 | | 1 |
| | *Nyctalus noctula* | 1 | | 1 |
| | *Pipistrellus kuhlii* | 1 | 2 | 3 |
| | *Pipistrellus nathusii* | | 1 | 1 |
| | *Pipistrellus pipistrellus* | | 7 | 7 |
| | *Pipistrellus pygmaeus* | | 1 | 1 |
| | *Plecotus auritus* | 1 | | 1 |
| | *Plecotus austriacus* | 2 | 1 | 3 |
| | *Vespertilio murinus* | | 2 | 2 |
| Rhinolophidae | *Rhinolophus ferrumequinum* | | 1 | 1 |
| | *Rhinolophus hipposideros* | | 2 | 2 |
| Miniopteridae | *Miniopterus schreibersii* | | 3 | 3 |
| | Total of samples tested | | | 37 |

*: identification based on morphological criteria.

## PCR: Amplification of the partial cytochrome b gene

Five μL of extracted DNA diluted to 1 ng/μL was used as template for amplification of a portion of the mitochondrial cyt b gene. We used the universal primers previously described in Lopez-Oceja et al. (2016) (forward primer L15601: 5′–TACGCAATCCTACGATCAATTCC–3′ and reverse primer H15748: 5′–GGTTGTCCTCCAATTCATGTTAG–3′) to amplify a 148 bp fragment of cyt b [27].

PCR amplification was performed in a 25 μL reaction volume containing 5 μL of DNA (1 ng/μL), 2.5 μL of 10X PCR Buffer without MgCl$_2$ (Invitrogen, Marseille, France), 1 μL of 50

**Table 3. Characteristics of the 31 bat faecal specimens included in the study.**

| Family | Bat species* | Number bat faecal specimens |
|---|---|---|
| | | 2019 |
| Vespertilionidae | *Eptesicus serotinus* | 1 |
| | *Myotis emarginatus* | 2 |
| | *Myotis myotis* | 22 |
| | *Pipistrellus pipistrellus* | 1 |
| | *Plecotus austriacus and/or Plecotus auritus* | 2 |
| Rhinolophidae | *Rhinolophus hipposideros* | 3 |
| | Total of samples tested | 31 |

*: identification based on morphological criteria.

mM MgCl$_2$, 1 μL of dNTPs (10 mM each) and 0.5 μL of Taq DNA polymerase (5 U) (Invitrogen, Marseille, France) and 1 μL of each primer (0.4 μM). The PCR was performed with the following conditions: 3 min at 95°C, 45 cycles of 30 s at 95°C, 30 s at 48°C and 45 s at 72°C and following with a final step of extension of 5 min at 72°C.

With each run, negative and positive PCR controls were performed for PCR validation.

## PCR: Amplification of partial D-loop

Five μL of extracted DNA was used for PCR amplification of the hypervariable domain II of the mtDNA control D-loop region producing PCR amplicons of 424 bp. The PCR amplifications were performed in 25 μL reaction volumes using validated primers described in Moussy et al. (2015). PCR consisted of an initial denaturation step at 95°C for 3 min, followed by 45 cycles of 95°C for 30 s, 56°C for 30 s, 72°C for 45 sec, and a final extension step of 72°C for 5 min. The 25 μL reaction mixture consisted of 25 μL of DNA template diluted to 1:10, 0.5 μL of enzyme mix in 2.5 μL 10X reaction buffer, 1.0 μL of 50 mM MgCl$_2$, 1 μL of 10 mM dNTP mixture, and 1 μM of each forward and reverse primer. The D-loop primers used for PCR amplification were L-strand D-loop (5′-CTACCTCCGTGAAACCAGCAAC-3′) and H-strand D-loop (5′-CGTACACGTATTCGTATGTATGTCCT-3). With each run, negative and positive PCR controls were performed for PCR validation. The D-loop PCR was performed on serotine bats (n = 5), only. The specificity of the PCR products was confirmed by direct sequencing of the amplified amplicons.

## Sequencing and phylogenetic analysis

Amplicons were analysed using 2% agarose gels stained with the intercalant SYBR Safe (Thermo Fisher Scientific, IIIkirch, France) then visualized using Bioimager (Bio-Rad, Roanne, France).

Sanger sequencing of PCR products was carried out by a service provider (Eurofins, Ebersberg, Germany) with the reverse and forward primers used in the PCR. All nucleotide sequences were assembled using Vector NTI software (version 11.5.3) (Invitrogen, France). Sequence alignments and determination of the percentages of identities and similarities were carried out with BioEdit Software (version 7.2.5) and MEGA X.

Genetic identification was determined using BLAST (Basic Local Alignment Search Tool) and by constructing a phylogenetic tree with MEGA-X using the maximum likelihood algorithm and the Tamura-Nei model between the 25 sequences from this study (representing 2 families and 15 species) and 52 representatives of bat species (3 families, 29 species) (Table 4). The bootstrap probabilities of each node were calculated using 500 replicates to assess the robustness of the maximum likelihood method. Bootstrap values over 70% were regarded as significant for phylogenetic analysis.

The nucleotide sequences were identified using BLASTN with the following parameters: standard nucleotide database and standard algorithm parameters by default (threshold of 0.05 and mismatch scores of 1,-2). In each case, the top BLAST hit was retained if the BLAST alignment covered more than 95% of the query length and the BLAST high-scoring segment pair identity was greater than ~90%.

## Results

### Genetic identification of bat carcasses and bat faeces

**Bat carcasses.**    Of 163 bat wing punches tested using cyt b PCR, 152 were genetically identified by BLAST analysis and/or phylogeny. The 152 genetically identified samples represented

**Table 4. Characteristics of the partial cytochrome b gene reference sequences retrieved from GenBank and other sequences amplified from wing punches and bat guano from this study.**

| No. | Country | Species | Year | GenBank Accession no. | Source |
|---|---|---|---|---|---|
| 1 | Japan | *Rhinolophus ferrumequinum* | 2003 | AB085730 | [32] |
| 2 | Japan | *Rhinolophus ferrumequinum* | 2003 | AB085731 | [32] |
| 3 | Japan | *Plecotus auritus* | 2003 | AB085734 | [32] |
| 4 | Japan | *Myotis daubentoni* | 2003 | AB106589 | [33] |
| 5 | Japan | *Myotis nattereri* | 2003 | AB106606 | [33] |
| 6 | Japan | *Vespertilio murinus* | 2010 | AB287358 | [34] |
| 7 | Swiss | *Nyctalus leisleri* | 2001 | AF376832 | [35] |
| 8 | Swiss | *Eptesicus nilssoni* | 2001 | AF376836 | [35] |
| 9 | Swiss | *Myotis blythii* | 2001 | AF376842 | [35] |
| 10 | Swiss | *Myotis bechsteinii* | 2001 | AF376843 | [35] |
| 11 | Swiss | *Myotis brandtii* | 2001 | AF376844 | [35] |
| 12 | Swiss | *Myotis capaccinii* | 2001 | AF376845 | [35] |
| 13 | Swiss | *Myotis dasycneme* | 2001 | AF376846 | [35] |
| 14 | Swiss | *Myotis emarginatus* | 2001 | AF376849 | [35] |
| 15 | Cyprus | *Pipistrellus pygmaeus* | 2004 | AJ504442 | [36] |
| 16 | Greece | *Pipistrellus pipistrellus* | 2004 | AJ504443 | [36] |
| 17 | Macedonia | *Pipistrellus kuhli* | 2004 | AJ504444 | [36] |
| 18 | Swiss | *Pipistrellus nathusii* | 2004 | AJ504446 | [36] |
| 19 | Swiss | *Hypsugo savii* | 2004 | AJ504450 | [36] |
| 20 | Swiss | *Myotis alcathoe* | 2004 | AJ841955 | [37] |
| 21 | Swiss | *Nyctalus noctula* | 2004 | AJ841967 | [37] |
| 22 | Spain | *Myotis myotis* | 2007 | AM261883 | [38] |
| 23 | China | *Myotis blythii* | 2006 | AM284170 | [39] |
| 24 | Japan | *Myotis daubentoni* | 2012 | AY665137 | [40] |
| 25 | Japan | *Myotis brandtii* | 2012 | AY665139 | [40] |
| 26 | Japan | *Plecotus auritus* | 2012 | AY665169 | [41] |
| 27 | China | *Miniopterus schreibersii* | 2004 | EF530339 | [41] |
| 28 | China | *Miniopterus schreibersii* | 2004 | EF530342 | [33] |
| 29 | China | *Plecotus auritus* | 2015 | EF570882 | [42] |
| 30 | Spain | *Rhinolophus euryale* | 2009 | EU436671 | [43] |
| 31 | Spain | *Rhinolophus mehelyi* | 2009 | EU436672 | [43] |
| 32 | Azerbaijan | *Eptesicus serotinus* | 2009 | EU751000 | [44] |
| 33 | Russia | *Eptesicus nilssoni* | 2009 | GQ272582 | [45] |
| 34 | Russia | *Eptesicus serotinus* | 2009 | GQ272585 | [45] |
| 35 | Russia | *Eptesicus serotinus* | 2009 | GQ272586 | [45] |
| 36 | Armenia | *Myotis myotis* | 2009 | GU817388 | [46] |
| 37 | France | *Myotis escalerai* | 2012 | JF412390 | [47] |
| 38 | France | *Myotis escalerai* | 2012 | JF412391 | [47] |
| 39 | France | *Myotis nattereri* | 2012 | JF412411 | [47] |
| 40 | Portugal | *Barbastella barbastellus* | 2012 | JQ683211 | [48] |
| 41 | Swiss | *Nyctalus leisleri* | 2012 | JX570901 | [49] |
| 42 | Greece | *Nyctalus noctula* | 2012 | JX570902 | [49] |
| 43 | France | *Rhinolophus hipposideros* | 2013 | KC978712 | [50] |
| 44 | Spain | *Rhinolophus mehelyi* | 2014 | KF031265 | [51] |
| 45 | Spain | *Rhinolophus mehelyi* | 2014 | KF031266 | [51] |
| 46 | France | *Rhinolophus euryale* | 2014 | KF031267 | [51] |

*(Continued)*

**Table 4.** (Continued)

| No. | Country | Species | Year | GenBank Accession no. | Source |
|---|---|---|---|---|---|
| 47 | France | *Rhinolophus euryale* | 2014 | KF031268 | [51] |
| 48 | Greece | *Myotis blythii* | 2013 | KF312501 | [52] |
| 49 | Iran | *Pipistrellus pipistrellus* | 2013 | KF874519 | [53] |
| 50 | Caucasus region | *Myotis mystacinus* | 2016 | KU060256 | [54] |
| 51 | Caucasus region | *Myotis mystacinus* | 2016 | KU060257 | [54] |
| 52 | Caucasus region | *Myotis alcathoe* | 2016 | KU060271 | [54] |
| 53 | France | *Barbastella barbastellus_132883* | 2018 | MZ066766 | This Study |
| 54 | France | *Eptesicus serotinus_133164* | 2019 | MZ066767 | This Study |
| 55 | France | *Nyctalus noctula_132681* | 2018 | MZ066769 | This Study |
| 56 | France | *Myotis mystacinus_133119* | 2019 | MZ066772 | This Study |
| 57 | France | *Myotis mystacinus_133333* | 2019 | MZ066774 | This Study |
| 58 | France | *Pipistresllus pipistrellus_133323* | 2019 | MZ066788 | This Study |
| 59 | France | *Myotis nattereri_133147* | 2019 | MZ066775 | This Study |
| 60 | France | *Nyctalus leisleri_132631* | 2018 | MZ066776 | This Study |
| 61 | France | *Nyctalus noctule_132624* | 2018 | MZ066777 | This Study |
| 62 | France | *Pipistrellus kuhli_133328* | 2019 | MZ066781 | This Study |
| 63 | France | *Plecotus auratus_132673* | 2018 | MZ066778 | This Study |
| 64 | France | *Plecotus austriacus_133165* | 2019 | MZ066779 | This Study |
| 65 | France | *Pipistrellus nathusius_133149* | 2019 | MZ066782 | This Study |
| 66 | France | *Pipistrellus pipistrellus_133225* | 2019 | MZ066773 | This Study |
| 67 | France | *Pipistrellus pipistrellus_133120* | 2019 | MZ066783 | This Study |
| 68 | France | *Pipistrellus pipistrellus_133152* | 2019 | MZ066784 | This Study |
| 69 | France | *Pipistrellus pipistrellus_133522* | 2019 | MZ066785 | This Study |
| 70 | France | *Pipistrellus pipistrellus_133330* | 2019 | MZ066786 | This Study |
| 71 | France | *Pipistrellus pipistrellus_133331* | 2019 | MZ066787 | This Study |
| 72 | France | *Myotis myotis_132714* | 2018 | MZ066770 | This Study |
| 73 | France | *Myotis emarginatus_Hoerdt* | 2019 | MZ066768 | This Study |
| 74 | France | *Plecotus austriacus_Weiler* | 2019 | MZ066780 | This Study |
| 74 | France | *Rhinolophus ferrumequinum_133127* | 2019 | MZ066789 | This Study |
| 76 | France | *Myotis myotis_GM-5-CB* | 2019 | MZ066771 | This Study |
| 77 | France | *Rhinolophus hipposideros_133128* | 2019 | MZ066790 | This Study |

the 3 families currently distributed throughout France with bat species belonging to the families Miniopteridae (n = 1), Rhinolophidae (n = 2) and Vespertilionidae (n = 19), respectively (Table 5). Twenty species out of the 35 bat species reported to date in France were genetically determined with an over representation of Pipistrelle bats in the sampling (37% = 61/163*100). BLAST analysis allowed the identification of 2 bat species belonging in the Rhinolophidae family with ~96% of nucleotide similarity with the GenBank sequences KU531352 (*R. hipposideros*) and MH029812 (*R. ferrumequinum*) and the identification of *M. schreibersii* from the Miniopteridae family with 93% of nucleotide similarity with the MK737740 sequence. Within, the Vespertilionidae family, 16 bat species were genetically identified by BLAST with a % nucleotide identity ranging from 87% to 100% (S1 Table).

Twenty out of the 156 samples belonging in the Vespertilionidae family could not be identified by BLAST sequence analysis of the cyt b amplicons. These samples had previously been morphologically determined as *E. serotinus* (n = 6), *V. murinus* (n = 2), *E. nilssonii* (n = 1), and Plecotus sp (n = 11). Interestingly, the phylogeny allowed the genetic determination of two

**Table 5. Results of PCR on the partial cytochrome b gene and species misidentification of bat wing punch samples compared with morphological species identification.**

| Family | Bat species * | Bat wing punches | | | Species |
|---|---|---|---|---|---|
| | | 2018–2019 | Morphological misidentification of bat species | Clarifications | |
| Vespertilionidae | *Barbastella barbastellus* | 1 | 0/1 | | |
| | *Eptesicus serotinus* | 8 | 0/8 | | |
| | *Eptesicus nilssonii* | 1 | 0/1 | | |
| | *Myotis bechsteinii* | 1 | 0/1 | | |
| | *Myotis brandtii* | 1 | 1/1 | | |
| | *Myotis daubentonii* | 3 | 1/3 | | |
| | *Myotis emarginatus* | 1 | 0/1 | | |
| | *Myotis myotis* | 11 | 2/11 | | |
| | *Myotis mystacinus* | 5 | 2/5 | | |
| | *Myotis nattereri* | 3 | 0/3 | | |
| | *Nyctalus leisleri* | 13 | 0/13 | | |
| | *Nyctalus noctula* | 10 | 1/10 | | |
| | *Pipistrellus kuhlii* | 7 | 2/7 | | |
| | *Pipistrellus nathusii* | 17 | 1/17 | | |
| | *Pipistrellus pipistrellus* | 25 | 3/25 | | |
| | *Pipistrellus pygmaeus* | 4 | 4/4 | | |
| | *Plecotus* | 1 | 0/1 | | |
| | *Plecotus auritus* | 7 | 2/7 | | |
| | *Plecotus austriacus* | 9 | 1/9 | | |
| | *Vespertilio murinus* | 2 | 0/2 | | |
| | *Pipistrellus* sp. | 8 | 0/8 | 8/8 | *Pp (n = 4), Pk (n = 4)* |
| | n.d. | 18 | - | 18/18 | *Pp (n = 6), Pk (n = 7), Rh (n = 1), Md (n = 1), Hs (n = 2), Pg/Pp (n = 1)* |
| Rhinolophidae | *Rhinolophus ferrumequinum* | 2 | 0/2 | | |
| | *Rhinolophus hipposideros* | 4 | 0/4 | | |
| Miniopteridae | *Miniopterus schreibersii* | 1 | 0/1 | | |
| | Total of samples tested | 163 | 21/163 | | |

*: identification based on morphological criteria.

**Abbreviations**: Pipistrellus pipistrellus (Pp), *Pipistrellus kuhlii (Pk)*, *Rhinolophus hipposideros (Rh)*, *Myotis daubentonii (Md)*, *Hypsugo savii (Hs)*, *Pipistrellus pygmaeus or Pipistrellus pipistrellus (Pg/Pp)*.

species, *Plecotus austriacus* and *Plecotus auritus* for 9 samples analysed with a boostrap of 99 (Fig 1).

The partial D-loop amplification (424-bp) of five bats morphologically identified as *E. serotinus* showed 100% of nucleotide similarity with *E. serotinus* (GenBank no. accession MF187797.1).

Of the 163 bat carcasses tested, 18 carcasses had not been previously identified with morphological criteria. The sequence analysis by BLAST and/or phylogeny showed for the 18 undetermined bats the following species: *P. pipistrellus (n = 6)*, *P. kuhlii (n = 7)*, *M. daubentonii (n = 1)*, *R. hipposideros (n = 1)*, *Hypsugo savii (n = 2)* and *P. pipistrellus or P.pygmaeus (n = 1)*.

**Bat faeces.** The analyses of cyt b sequences led to a specific identification of the 31 samples of bat species from one faecal pellet for the seven bat species tested (Table 6).

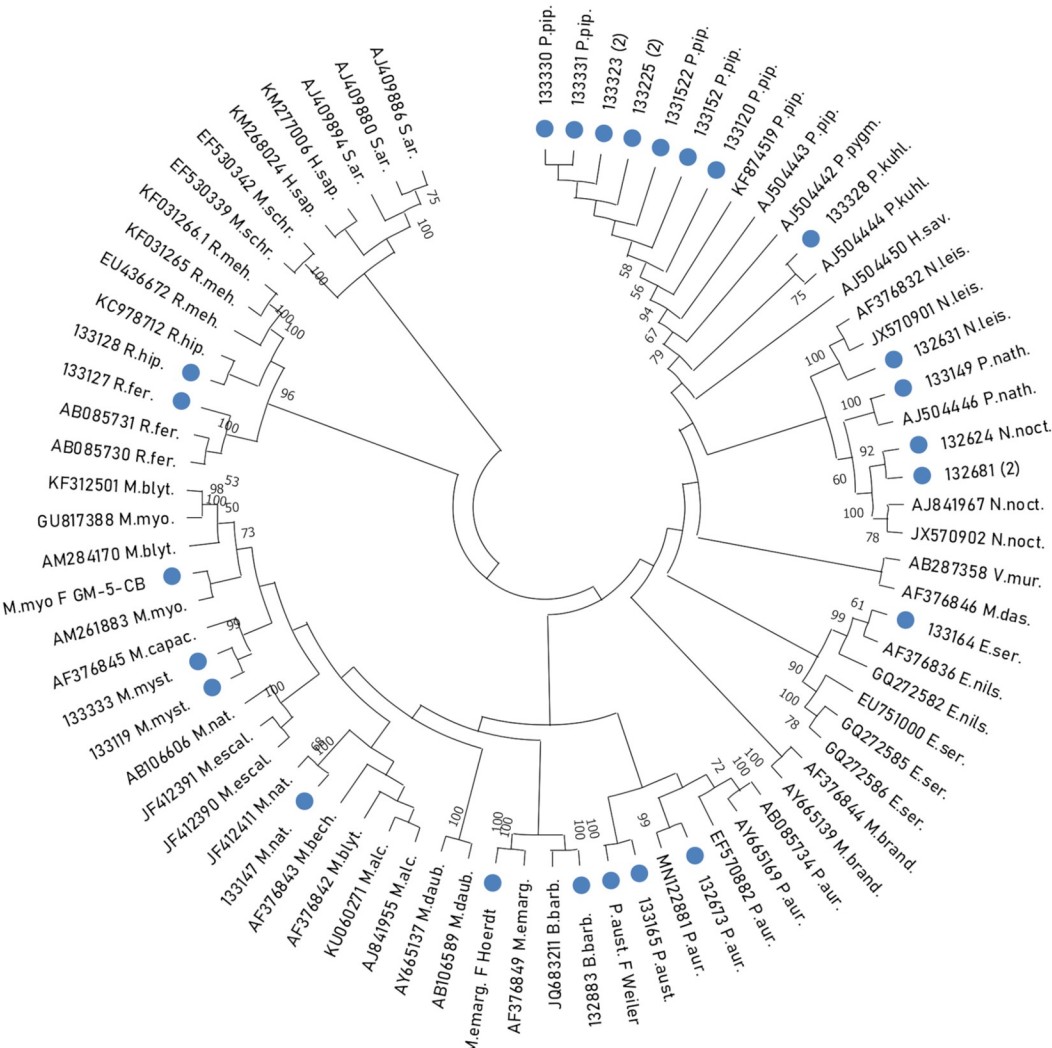

**Fig 1. Phylogenetic tree of the partial cytochrome b (cyt b) gene of 52 referenced sequences and 25 bat sequences representing 15 autochthonous bat species. B.barb**: *Barbastella barbastellus*, **E.nils**: *Eptesicus nilssonii*, **E.ser**: *Eptesicus serotinus*, **H.s**: *Homo sapiens*, **H.savi**: *Hypsugo savii*, **M.al**: *Myotis alcathoe*, **M.bech**: *Myotis bechsteinii*, **M.bly**: *Myotis blythii*, **M.br**: *Myotis brandtii*, **M.c**: *Myotis capaccinii*, **M.daub**: *Myotis daubentonii*, **M.das**: *Myotis dasycneme*, **M.em**: *Myotis emarginatus*, **M.esc**: *Myotis escalerai*, **M.myo**: Myotis *Myotis*, **M.mys**: *Myotis mystacinus*, **M. nat**: *Myotis nattereri*, **M.schr**: *Myotis schreibersii*, **N.leis**: *Nyctalus leisleri*, **N.noct**: *Nyctalus noctula*, **P.aur**: *Plecotus auritus*, **P.aust**: *Plecotus austriacus*, **P. kuh**: *Pipistrellus kuhlii*, **P.pip**: *Pipistrellus pipistrellus*, **P.pyg**: *Pipistrellus pygmaeus*, **P.nath**: *Pipistrellus nathusii*, **R.fer**: *Rhinolophus ferrumequinum*, **R.hip**: *Rhinolophus hipposideros*, **R.meh**: *Rhinolophus mehelyi*, **S.a**: *Sorex araneus*, **V.mur**: *Vespertilio murinus*.

The 31 genetically identified samples represented 2 out of the 3 families currently distributed throughout France with bat species belonging to the families Rhinolophidae (n = 1) and Vespertilionidae (n = 3), respectively (Table 6).

BLAST analysis allowed the identification of the bat species, *R. hipposideros* with ~96% of nucleotide similarity with the GenBank KU531352 and KC978344 sequences. ~ 94% of similarity were shown between bats morphologically identified as *M. emarginatus* and the AF376849 GenBank sequence representative of *M. emarginatus*. Within the two species *P.*

**Table 6. Results of PCR on the partial cytochrome b gene and species misidentification of bat guano samples compared with morphological identification.**

| Family | Bat species* | Nb. bat faecal specimens 2019 | Morphological misidentification of bat species | Clarifications | Species |
|---|---|---|---|---|---|
| Vespertilionidae | *Eptesicus serotinus* | 1 | 1/1 | | Rh (n = 1) |
| | *Myotis emarginatus* | 3 | 0/2 | | |
| | *Myotis myotis* | 20 | 0/22 | | |
| | *Pipistrellus pipistrellus* | 1 | 0/1 | | |
| | *Plecotus auritus* and/or *Plecotus austriacus* | 2 | 1/2 | | Rh (n = 1) |
| Rhinolophidae | *Rhinolophus hipposideros* | 3 | 0/3 | | |
| | Total of samples tested | 32 | 0/31 | | |

\*: identification based on morphological criteria.

**Abbreviations**: *Rhinolophus hipposideros (Rh)*.

*pipistrellus* and *M. myotis*, a mean of 99% were shown between the faecal samples and the Genbank sequences KJ765388.1 (*M. myotis*) and AH006588.2 (*P. pipistrellus*).

Interestingly, and as for bat carcasses, the samples that had previously been morphologically determined as *Plecotus* sp (n = 11) could not be identified by BLAST sequence analysis of the cyt b amplicons but was identified by phylogeny with a bootstrap of 99 (Fig 1).

## Comparison between morphological identification and cyt b PCR analysis

**Bat carcasses.** The comparison between morphological and genetic identification carried out on the 163 bat samples showed the same results for 142 samples tested and identified a total of 21 misidentifications. These 21 morphological misidentifications represented ~12.5% of total bat carcasses tested. The misidentifications were confirmed by bat specialists who performed a second morphological identification on these bat samples using another species determination key. Results are detailed in Table 4. Morphological identification errors were reported for 11 species: *M. brandtii* (n = 1), *M. daubentonii* (n = 1), *M. myotis* (n = 2), *M. mystacinus* (n = 2), *N. noctula* (n = 1), *P. kuhlii* (n = 2), *P. nathusii* (n = 1), *P. pipistrellus* (n = 3), *P. pygmaeus* (n = 5), *P. auritus* (n = 2) and *P. austriacus* (n = 1) (S1 Table).

Genetic identification allowed clarifications for 26 bats tested (18 bats morphologically identified as not determined and 8 bats morphologically identified as *Pipistrellus* sp.) (Table 5).

BLAST analysis allowed the distinction of the 8 Pipistrelle bats tested in the study with the genetic identification of 4 *P. pipistrellus* (~96% similarity with the Genbank AH006588.2 and AJ504443.1 sequences) and 4 *P. Kuhlii* (95% similarity with the Genbank MN045571.1 sequence representative of *P. Kuhlii*). Of the 18 bat carcasses morphologically not determined, we reported the identification of 6 species belonging in the Vespertilionidae family with *P. pipistrellus (n = 6), P. kuhlii (n = 7), M. daubentonii (n = 1), R. hipposideros (n = 1), Hypsugo savii (n = 2)* and *P. pipistrellus* or *P. pygmaeus (n = 1)*.

**Bat faeces.** The genetic identification of bat species from the guano samples showed 2 morphological misidentifications out of the 31 guano samples tested. Misidentifications were reported in two sites: the site 22 among *Plecotus sp.* and *R. hipposideros* and the site 31 among *E. serotinus* and *R. hipposideros* (S1 Table).

## Discussion and conclusion

To our knowledge, this study is the first to evaluate published universal primers targeting the cyt b gene [27] from two different bat matrices, wing punch and guano, to genetically identify autochthonous bat species. Of the 35 species reported in France, 14 species are uncommon in France. Our study showed 12.5% of misidentification for 11 out of the 22 bat species tested. Our results corroborate the Nadin-Davis (2012) study, which also showed non-negligible percentages of morphological bat species misidentification of between 10 and 15%.

It is rare and very complicated to collect samples for research or rabies diagnosis from autochthonous bats. The fact that all bat carcasses included in this study came from a sample collection compiled for rabies diagnosis at ANSES Laboratory led to an over representation of P. pipistrellus in our sampling. In France, *P. pipistrellus* is a very common bat species compared with other bat species. On average, there is one *P. pipistrellus* colony in each town in France (Laurent Arthur, personal communication). *P. pipistrellus* represents on average between 45 and 50% of the total number of carcasses in the rabies diagnosis sample collection. In our study, *P. pipistrellus* represented 16% of the total number of samples.

The species could not be identified for 11 of the 163 samples tested. These samples were morphologically identified as *E. serotinus* (n = 6), *E. nilssonii* (n = 1), *V. murinus* (n = 2) and *Plecotus sp (n = 2)*. One hypothesis of species non-identification is that the cyt b PCR was not able to identify these 8 samples due to DNA degradation. Two published studies investigated the genetic structure of *E. serotinus* bats by amplifying the partial D-loop region [25, 26]. Thus, the amplification of the partial D-loop region on the five *E. serotinus* was successful and our results on Sanger sequencing confirmed the morphological species determination as *E. serotinus*.

Regarding bat faecal specimens, results and analyses of the 31 amplicons showed that the cyt b PCR allowed specific identification of bat species from just one faecal pellet of bat guano. Bat species have previously been genetically identified from guano samples by amplification of a segment of the cox1 mitochondrial gene using real-time PCR [21]. Some studies have demonstrated the advantages of using real-time PCR compared with conventional PCR: real-time PCR is more sensitive, specific and rapid as a diagnostic method for detecting *Vibrio vulnificus* and *Samonella* spp. compared with conventional PCR [55, 56]. Both PCR techniques are equally effective for detection of the genome of visceral leishmaniasis [57]. The discrepancy between the results obtained in our study and those of the Walker et al. study likely arises from using a traditional PCR with the cyt b gene universal primers [21, 27]. In our study, the genetic determination of bats was based on universal primers of the cytb gene, described by Lopez-Oceja et al., as highly specific, especially for highly degraded DNA samples (Lopez-Oceja et al., 2016). Species identification from bat faecal samples can also be undertaken by DNA mini-barcode assay based on the amplification of a segment of the mitochondrial gene cytochrome c oxidase I (COI) [21]. New primers targeting a 580 bp fragment of the COI gene were described for the identification of bat species [21]. Interestingly, the comparison between the cytb and COI genes was studied by Tobe et al. for reconstructing mammalian phylogenies [58]. Their results tend to support the use of Cytb over that of COI. Conventional PCR allowed us to obtain nucleotide sequences from amplicons and to genetically determine bat species using BLAST and/or phylogeny. In addition, the cost of real-time PCR is higher than conventional PCR. In our study, we demonstrated the efficacy of using universal cyt b primers to genetically identify autochthonous bats from faecal samples, a non-invasive method.

The cyt b PCR made it possible to determine 18 bat samples that could not initially be identified based on morphological criteria. Non-determination of bats can be attributed to the state of decomposition of bat carcasses, the age of the bat, especially for juveniles or pups, or

inexperienced bat naturalists. Morphological identification of bat species is usually carried out on living bats. Some morphological features disappear if the carcasses are not fresh, and identification becomes more complicated, creating a source of errors [59, 60].

It is important to identify bat species to preserve bats, which play a key role in the environment. Bats play an important biological and ecological role and many studies have suggested that they are reservoirs in the transmission of many zoonoses and infectious diseases from animals to humans [3, 9, 61]. To better understand bats and their role in the circulation of pathogens, specific and precise identification of bat species is required. Our results here showed that genetic identification is an efficient way to identify bat species in France and is a rapid and reliable tool to use compared with morphological identification.

## Supporting information

**S1 Table. Raw data analysis: Bat identification from bat wing punches and faecal samples.**
(XLSX)

**S1 Raw images.**
(TIF)

## Acknowledgments

We thank our collaborators and bat naturalists for providing the bat specimens used in this study. We also thank Alexis Petitdemange, Mélanie Badré Biarnais and Jean Luc Schereffer for their technical assistance in this study.

## Author Contributions

**Conceptualization:** Zouheira Djelouadji, Evelyne Picard-Meyer.

**Formal analysis:** Youssef Arnaout, Zouheira Djelouadji, Emmanuelle Robardet, Julien Cappelle.

**Investigation:** Youssef Arnaout, Zouheira Djelouadji, Evelyne Picard-Meyer.

**Methodology:** Youssef Arnaout, Evelyne Picard-Meyer.

**Project administration:** Zouheira Djelouadji.

**Supervision:** Zouheira Djelouadji, Evelyne Picard-Meyer.

**Validation:** Zouheira Djelouadji, Emmanuelle Robardet, Julien Cappelle, Florence Cliquet, Frédéric Touzalin, Giacomo Jimenez, Suzel Hurstel, Christophe Borel, Evelyne Picard-Meyer.

**Writing – original draft:** Youssef Arnaout, Zouheira Djelouadji, Emmanuelle Robardet, Julien Cappelle, Florence Cliquet, Frédéric Touzalin, Giacomo Jimenez, Suzel Hurstel, Christophe Borel, Evelyne Picard-Meyer.

**Writing – review & editing:** Youssef Arnaout, Zouheira Djelouadji, Emmanuelle Robardet, Julien Cappelle, Florence Cliquet, Frédéric Touzalin, Giacomo Jimenez, Suzel Hurstel, Christophe Borel, Evelyne Picard-Meyer.

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
