## [Decision Letter · Decision Letter 0]

1 Jul 2021

PONE-D-21-14577

Genetic identification of bat species for pathogens surveillance across France

PLOS ONE

Dear Dr. Picard-Meyer,

Thank you for submitting your manuscript to PLOS ONE. After careful consideration, we feel that it has merit but does not fully meet PLOS ONE’s publication criteria as it currently stands. Therefore, we invite you to submit a revised version of the manuscript that addresses the points raised during the review process.

We look forward to receiving your revised manuscript.

Kind regards,

Camille Lebarbenchon

Academic Editor

PLOS ONE

Reviewers' comments:

Reviewer's Responses to Questions

**Comments to the Author**

1. Is the manuscript technically sound, and do the data support the conclusions?

Reviewer #1: Yes

Reviewer #2: Yes

2. Has the statistical analysis been performed appropriately and rigorously? 

Reviewer #1: Yes

Reviewer #2: Yes

3. Have the authors made all data underlying the findings in their manuscript fully available?

Reviewer #1: Yes

Reviewer #2: Yes

4. Is the manuscript presented in an intelligible fashion and written in standard English?

Reviewer #1: Yes

Reviewer #2: Yes

5. Review Comments to the Author

Reviewer #1: This study describes the test of previously published primers for the barcoding of french bat species, using both wing punch from bat carcasses and faecal pellets. Results showed that the method is successful in identifying at least 22 from the 35 bat species described in France. This technique allows to correct 12.5% of morphological misidentifications among 167 samples, and identify 24 samples that could not be morphologically identify in the field. This study provides new and important tools for the correct identification of bat species, which is of primary importance for ecological and pathogen-associated studies of bats.

I enjoyed reading this manuscript which is globally well written. However, I recommend the authors to add the line numbers on the manuscript to help the reviewing process. My two major concern are the lack of information for the phylogenetic analysis and D-loop PCR. The authors should better explain why they perform the phylogenetic analysis in addition to the BLAST, and better present the results of the grouping in the tree. Also throughout the manuscript, it is not clear why the authors also performed PCR on the D-loop, but only for 1 bat species and on a very limited number of samples. Please see below for more detailed comments.

1) Page 1, title: Please correct “pathogens surveillance” to “pathogen surveillance”.

2) Page 5: please correct ‘bats species’ to ‘bat species’.

3) Methods, page 6: were the faecal pellets collected directly from bat handling or from the environment?

4) Methods, sequencing and phylogenetic analysis: please provide more details on which criteria were used in the BLAST output to identify bat species? And for the phylogenetic tree analysis? A better explanation of phylogenetic results would also be valuable, for example, whether or not the tested samples clustered with previous reference sequences (=are the species-clusters are well supported or not). Did you assume correct identification if the sample clustered with reference sequences?

5) Page 11: genetic distances were calculated but I did not see the results of this analysis.

6) Results, page 12: please remove ‘by’ in ‘by followed…’

7) Results, page 12: what do the authors mean by ‘optimization’? I don’t see any description of the(PCR?) optimization in the methods. I don’t understand if the protocol was different for the 37 samples used for optimization. Please provide more details in the methods and results.

8) Why only serotine bats were tested for D-loop and only 6 individuals? I understood later from the discussion that the D-loop PCR was performed because there were no amplification with the cytb, and this allowed confirming the ID for E. serotinus. But why the D-loop PCR was not done for the other non-cytb amplified samples (one E. nilsonni and one V. murinus)?

9) Page 12: Pl. auritus should be P. auritus.

10) Results, page 12: Genetic identification of bat carcasses: ‘the panel of genetically…M. schreibersii (n=1)’. It looks like that the authors give here the results of the cytb genetic identification. Is it not just the listing of the samples used in this analysis (same as in table 1A). If so, this is a bit redundant with information presented in the methods. I suggest to remove this sentence and present directly the comparison of morphological/genetic identification.

11) Figure 3: It is not clear what are the sequences produced in this study. Please used bold or color to highlight them. Please also add posterior values at the nodes. Is the tree well supported? It may also be clearer if the different bat species were better delineated (using colored boxes for example).

12) For the 25 sequences included in the tree, was the BLAST results the same as the tree classification? Why only 25 sequences were included in the phylogenetic tree, and not all the sequences produced (n=167)? Why all these sequences were not submitted to Genbank?

13) Page 14: ‘Sequences analyses using BLAST… following species…’. I don’t understand this result. Authors mentioned above that errors of morphological determination were observed for 11 species.

14) Page 13: Genetic identification of bat faeces: ‘1 was not determined’. Does this mean the BLAST and phylogeny analyses did not give any conclusive results?

15) Discussion, second paragraph: ’It is rare…’ I’m not sure to understand the relevance of this paragraph. Is this to justify the non-homogeneity of number of samples per bat species? If so, I would just state that : “the fact that all bat carcasses…ANSES Laboratory, leading to an over representation of P. pipistrellus in our sampling’.

16) Page 21: please give details on what is the discrepancy between the present study and that of Walker’s, to make it clearer for non-specialist readers.

17) Page 22: the authors suggest that the non-amplification of the cytb for 8 samples (including E. serotinus, E. nilssonii and V. murinus) could be due to the short length of the PCR fragment. Why exactly? But they were successful in amplifying these 3 species for other samples and for the same gene fragment. I would rather suggest that the non-amplification results from degraded DNA. It would have been valuable to test the integrity of DNA by gel electrophoresis.

Reviewer #2: This study identified bats in France from wing punches and fecal material. The results are straightforward and a useful contribution to the identification of bat remains in France. I have made a number of minor edits and comments on the pdf as notes/sticky notes/deletions.

---

## [Author Response · Author response to Decision Letter 0]

5 Aug 2021

See below the responses to the remarks given by the two reviewers:

Corrections for the paper entitled “Genetic identification of bat species for pathogens surveillance across France” by Y. Arnaoult et al.

General comments:

The manuscript was completely revised according to the general and the specific comments of the two reviewers. Corrections and clarifications were given all along the manuscript, as suggested. 

A supplementary Table (S Table 1) was added to follow the remarks of the reviewer N°1. The two figures 1 and 2 were deleted to follow the reviewing remarks. The figure 3, renumbered 1, was slightly modified by the adding of a blue colored dot (differentiation between the Genbank sequences and the sequences of this study).

Responses to the remarks given by the two reviewers are written in blue following each question/remark. 

Following the e-mail dated of 19 July 2021, the three tables 3-5 were renumbered, as follows:

- Table 3 was renumbered Table 4 (See Page 20)

- Table 4 was renumbered Table 5 (See Page 21)

- Table 5 was renumbered Table 3 (See Page 13-14).

The three tables are refered in the text.

Review Comments to the Author

Reviewer 1:

Reviewer #1: This study describes the test of previously published primers for the barcoding of french bat species, using both wing punch from bat carcasses and faecal pellets. Results showed that the method is successful in identifying at least 22 from the 35 bat species described in France. This technique allows to correct 12.5% of morphological misidentifications among 167 samples, and identify 24 samples that could not be morphologically identify in the field. This study provides new and important tools for the correct identification of bat species, which is of primary importance for ecological and pathogen-associated studies of bats.

I enjoyed reading this manuscript which is globally well written. However, I recommend the authors to add the line numbers on the manuscript to help the reviewing process. My two major concern are the lack of information for the phylogenetic analysis and D-loop PCR. The authors should better explain why they perform the phylogenetic analysis in addition to the BLAST, and better present the results of the grouping in the tree. Also throughout the manuscript, it is not clear why the authors also performed PCR on the D-loop, but only for 1 bat species and on a very limited number of samples. Please see below for more detailed comments.

1) Page 1, title: Please correct “pathogens surveillance” to “pathogen surveillance”.

The remark has been taken into consideration. We changed the title and correct “pathogens surveillance” to “pathogen surveillance”. See Page 1, line 1.

2) Page 5: please correct ‘bats species’ to ‘bat species’.

To follow the remark, we correct ‘bats species’ to ‘bat species’. See Page 5, line 106.

3) Methods, page 6: were the faecal pellets collected directly from bat handling or from the environment?

The faecal pellets were collected directly from the environment. The sampling was carried out by naturalists who are participating each year in the monitoring of bat population in the Grand Est region in France.

We clarified the manuscript (See page 6, line 131-133): “Faecal pellets were collected directly from the environment in three different sites in the Grand Est region in France.”

4) Methods, sequencing and phylogenetic analysis: please provide more details on which criteria were used in the BLAST output to identify bat species? And for the phylogenetic tree analysis? A better explanation of phylogenetic results would also be valuable, for example, whether or not the tested samples clustered with previous reference sequences (=are the species-clusters are well supported or not). Did you assume correct identification if the sample clustered with reference sequences?

We agree with the remark of the reviewer and we clarified the manuscript by adding the criteria used to identify bat species in the BLASTN analysis. See page 12, lines 214-217. 

The following sentence was added: “The nucleotide sequences were identified using BLASTN with the following parameters: standard nucleotide database and standard algorithm parameters by default (threshold of 0.05 and mismatch scores of 1,-2). In each case, the top BLAST hit was retained if the BLAST alignment covered more than 95% of the query length and the BLAST high-scoring segment pair identity was greater than �90%.”

We have also clarified the phylogenetic analysis by detailing in the methods the calcul of bootstrap probabilities of each node to assess the robustness of the phylogeny tree. See page 6 lines 2010-213.

The following sentence was added: The bootstrap probabilities of each node were calculated using 500 replicates to assess the robustness of the maximum likelihood method. Bootstrap values over 70% were regarded as significant for phylogenetic analysis.

We also detailed in the results a deeper analysis of the phylogeny for each species. See the results section.

5) Page 11: genetic distances were calculated but I did not see the results of this analysis.

We deleted the mention “calculate the distance” as these results are not presented in this paper that are based on the BLAST analysis and the phylogenetic tree for the genetic identification of bat species.

6) Results, page 12: please remove ‘by’ in ‘by followed…’

The sentence was deleted to follow the remark of the reviewer 2.

7) Results, page 12: what do the authors mean by ‘optimization’? I don’t see any description of the(PCR?) optimization in the methods. I don’t understand if the protocol was different for the 37 samples used for optimization. Please provide more details in the methods and results.

This sentence was deleted to follow the remark of the reviewer 2. To answer to the query, the amplification of the partial cytB described by Lopez-Oceja et al., was slightly modified in the laboratory and validated against these 37 samples for determining the diagnosis sensitivity of the PCR. The method developed in the laboratory has been fully described in the Methods, in Page 10. 

8) Why only serotine bats were tested for D-loop and only 6 individuals? I understood later from the discussion that the D-loop PCR was performed because there were no amplification with the cytb, and this allowed confirming the ID for E. serotinus. But why the D-loop PCR was not done for the other non-cytb amplified samples (one E. nilsonni and one V. murinus)?

The remark of the reviewer is relevant. Indeed, it should have been interesting to check the specificity of the primers used for amplifying the D-loop region with V. murinus and/or E. nilssoni. However, the PCR was not done, because the PCR described by Moussy et al. was been tested on Eptesicus serotinus, only, and described as specific of this species.

9) Page 12: Pl. auritus should be P. auritus.

The section (lines 254 - 259) was deleted to follow the remark 10 (See Page 16).

10) Results, page 12: Genetic identification of bat carcasses: ‘the panel of genetically…M. schreibersii (n=1)’. It looks like that the authors give here the results of the cytb genetic identification. Is it not just the listing of the samples used in this analysis (same as in table 1A). If so, this is a bit redundant with information presented in the methods. I suggest to remove this sentence and present directly the comparison of morphological/genetic identification.

To follow the remark, we deleted the sentence. 

11) Figure 3: It is not clear what are the sequences produced in this study. Please used bold or color to highlight them. Please also add posterior values at the nodes. Is the tree well supported? It may also be clearer if the different bat species were better delineated (using colored boxes for example).

We agree with the remark of the reviewer. The figure 3 was slightly modified by differentiating the Genbank sequences from the sequences of this study. Before each bat sequence of the study, was added a blue colored dot. See Figure 1. (To follow the remark of the reviewer 2, the two figures 1 and 2 were deleted. Fig 3 is so renumbered as Fig 1 in the new version of the manuscript). 

12) For the 25 sequences included in the tree, was the BLAST results the same as the tree classification? Why only 25 sequences were included in the phylogenetic tree, and not all the sequences produced (n=167)? Why all these sequences were not submitted to Genbank?

13) Page 14: ‘Sequences analyses using BLAST… following species…’. I don’t understand this result. Authors mentioned above that errors of morphological determination were observed for 11 species.

We clarified the manuscript following the remarks 4, 12 and 13. 

To answer to the different remarks and in particular the remark 12 we included in the phylogenetic analysis only 1 sequence representative of each bat family, in order to reduce the redundancy of sequences.

The manuscript was clarified accordingly to the remarks with the supplementary Table 1 detailing the BLAST results and phylogeny for all samples. Phylogeny results were added in the S. Table 1 for the 25 samples as well as for the samples representing Plecotus auritus and P. austriacus, analysed by phylogeny. The supplementary Table 1 describes the genetic determination for the two types of matrices tested: faecal and bat wing punches. 

The section “Genetic identification of bat carcasses and bat faeces “ was modified to take account the remark. We added in this section, for both bat carcasses and bat faeces, the results of blast analysis for the 3 families, Rhinolophidae, Miniopteridae and Vespertilionidae.

14) Page 13: Genetic identification of bat faeces: ‘1 was not determined’. Does this mean the BLAST and phylogeny analyses did not give any conclusive results?

The mention “not determined” was deleted in the two tables 2 (page 9) and 5 (page 21). After checking the raws datas, there are” no undetermined” in the sampling.

15) Discussion, second paragraph: ’It is rare…’ I’m not sure to understand the relevance of this paragraph. Is this to justify the non-homogeneity of number of samples per bat species? If so, I would just state that : “the fact that all bat carcasses…ANSES Laboratory, leading to an over representation of P. pipistrellus in our sampling’.

The remark 15 has been taken into consideration. We changed the sentence by the following sentence: “It is rare and very complicated to collect samples for research or rabies diagnosis from autochthonous bats. The fact that all bat carcasses included in this study came from a sample collection compiled for rabies diagnosis at ANSES Laboratory led to an over representation of P. pipistrellus in our sampling.. In France, this bat species, P. pipistrellus is a very common bat species compared with other bat species…..”See page 27, lines 397 to 403. 

16) Page 21: please give details on what is the discrepancy between the present study and that of Walker’s, to make it clearer for non-specialist readers.

The remark has been taken into consideration and we have added in Discussion the following sentence:

In our study, the genetic determination of bats was based on universal primers of the cytb gene, described by Lopez-Oceja et al., as highly specific, especially for highly degraded DNA samples (Lopez-Oceja et al., 2016). Species identification from bat faecal samples can also be undertaken by DNA mini-barcode assay based on the amplification of a segment of the mitochondrial gene cytochrome c oxidase I (COI) (Walker et al., 2016). New primers targeting a 580 bp fragment of the COI gene were described for the identification of bat species (Walker et al., 2016). Interestingly, the comparison between the cytb and COI genes was studied by Tobe et al. for reconstructing mammalian phylogenies (Tobe et al., 2010). Their results tend to support the use of Cytb over that of COI. See page 28, lines 424-431.

17) Page 22: the authors suggest that the non-amplification of the cytb for 8 samples (including E. serotinus, E. nilssonii and V. murinus) could be due to the short length of the PCR fragment. Why exactly? But they were successful in amplifying these 3 species for other samples and for the same gene fragment. I would rather suggest that the non-amplification results from degraded DNA. It would have been valuable to test the integrity of DNA by gel electrophoresis.

To follow the remark, we changed the sentence. See page 27, lines 409 to 412. 

Reviewer 2:

Reviewer #2: This study identified bats in France from wing punches and fecal material. The results are straightforward and a useful contribution to the identification of bat remains in France. I have made a number of minor edits and comments on the pdf as notes/sticky notes/deletions.

Abstract: 

The abstract has been modified. See Page 1, line 22, 25 and 30.

Introduction:

We have changed the introduction, as suggested by the reviewer as follows: 

- “in” has been changed by “occur in”. See See Page 3, line 59.

- “1980s” has been changed by “the 1980s”. See See Page 4, line 75.

- “Since 1989” has been changed by “For example, since 1989” See See Page 4, line 75.

Results:

The two sentences have been deleted to follow the remark. See Page 12 Lines 220-228.

We added in Methods the sentence on the negative and positive controls that were well done for each run. See Page 11, line 186 and lines 196-197.

Genetic identification of bat faeces:

As suggested, we deleted the mention of the optimization of PCR in lines 266-268 and moved the sentence in the Materials section, in Page 6, lines 126-128.

Figure 3:

We agree with the remark of the reviewer. The figure 3 (renumbered Figure 1) was slightly modified by differentiating the Genbank sequences from the sequences of this study. Before each bat sequence of the study, was added a blue colored dot. 

Figure 1 and Figure 2:

As suggested, the two figures were deleted. Fig 3 has been renumbered as Fig 1 in the new version of the manuscript. 

Discussion and conclusion:

As suggested, we clarified the sentence by adding in Page 27, in line 392, “wing punch and guano”.

We deleted the sentence “This frequency can be explained ….vegetation” as well as the associated reference. See Page 27, lines 406 to 408.

In Page 29, line 442, we added as suggested by the reviewer, the two references (60,61), as follows:

- 59. Korstian JM, Hale AM, Bennett VJ, Williams DA. Using DNA barcoding to improve bat carcass identification at wind farms in the United States. Conserv Genet Resour. 1 mars 2016;8(1):27‑34. 

- 60. Chipps AS, Hale AM, Weaver SP, Williams DA. Genetic diversity, population structure, and effective population size in two yellow bat species in south Texas. PeerJ. 2020;8:e10348. 

- 

In Page 28, line 414, we changed “six” by “five”.

---

## [Decision Letter · Decision Letter 1]

1 Dec 2021

Genetic identification of bat species for pathogen surveillance across France

PONE-D-21-14577R1

Dear Dr. Picard-Meyer,

We’re pleased to inform you that your manuscript has been judged scientifically suitable for publication and will be formally accepted for publication once it meets all outstanding technical requirements.

Kind regards,

Daniel Becker

Academic Editor

PLOS ONE

Additional Editor Comments (optional):

Reviewers' comments:

Reviewer's Responses to Questions

**Comments to the Author**

1. If the authors have adequately addressed your comments raised in a previous round of review and you feel that this manuscript is now acceptable for publication, you may indicate that here to bypass the “Comments to the Author” section, enter your conflict of interest statement in the “Confidential to Editor” section, and submit your "Accept" recommendation.

Reviewer #1: All comments have been addressed

Reviewer #2: (No Response)

2. Is the manuscript technically sound, and do the data support the conclusions?

Reviewer #1: Yes

Reviewer #2: Yes

3. Has the statistical analysis been performed appropriately and rigorously? 

Reviewer #1: Yes

Reviewer #2: Yes

4. Have the authors made all data underlying the findings in their manuscript fully available?

Reviewer #1: Yes

Reviewer #2: Yes

5. Is the manuscript presented in an intelligible fashion and written in standard English?

Reviewer #1: Yes

Reviewer #2: Yes

6. Review Comments to the Author

Reviewer #1: The authors have responded to almost all the comments, but there is still incomplete information in the results and some rephrasement to do. Please see my comments below:

1) Line 23 : I think “herpes virus” should be “herpesvirus”.

2) Line 82: “pygmaeus and P. nathusii, or to…”: “and” should not be in italics.

3) Line 131: it is still not clear how the faecal pellets were associated to a bat species, as no capture was done. I guess that the pellets were collected under a monospecific bat colony, and that the bat species was determined by inspected hanging individuals in the colony? Also, how fresh were the pellets ? From the day, or probably several days, weeks ? Were they collected directly on the ground, or some plastic sheets were used ? Might be good to add these details.

4) Table 2 “n.d” in footnotes but not seen in the table.

5) Line 195: please correct “…TCCT-3). With each run,…”

6) Line 196: please explain in the text why only serotine bats were amplified with the Dloop.

7) Please delete “P. pipistrellus”, because of a repetition with “this bat species”.

Reviewer #2: Only one edit: Replace reference - Chipps AS, Hale AM, Weaver SP, Williams DA. Genetic diversity, population

structure, and effective population size in two yellow bat species in south Texas. PeerJ. 2020;8:e10348.

WITH

Chipps AS, AM Hale, SP Weaver, and DA Williams. 2020. Genetic approaches are necessary to accurately understand bat‐wind turbine impacts. Diversity 12:236.

7. PLOS authors have the option to publish the peer review history of their article (what does this mean?). If published, this will include your full peer review and any attached files.

Reviewer #1: No

Reviewer #2: No

---

## [Editor Report · Acceptance letter]

15 Dec 2021

PONE-D-21-14577R1 

Genetic identification of bat species for pathogen surveillance across France 

Dear Dr. Picard-Meyer:

I'm pleased to inform you that your manuscript has been deemed suitable for publication in PLOS ONE. Congratulations! Your manuscript is now with our production department. 

Kind regards, 

on behalf of

Dr. Daniel Becker 

Academic Editor

PLOS ONE